# Assessment of the Performance of Lactate Dehydrogenase-Based Rapid Diagnostic Test for Malaria in Djibouti in 2022–2023

**DOI:** 10.3390/diagnostics14030262

**Published:** 2024-01-25

**Authors:** Rahma Abdi Moussa, Nasserdine Papa Mze, Houssein Yonis Arreh, Aicha Abdillahi Hamoud, Kahiya Mohamed Alaleh, Fatouma Mohamed Aden, Abdoul-Razak Yonis Omar, Warsama Osman Abdi, Samatar Kayad Guelleh, Abdoul-Ilah Ahmed Abdi, Leonardo K. Basco, Bouh Abdi Khaireh, Hervé Bogreau

**Affiliations:** 1Aix Marseille Université, IRD, AP-HM, SSA, VITROME, 13005 Marseille, Francelkbasco@yahoo.fr (L.K.B.); 2IHU-Méditerranée Infection, 13005 Marseille, France; 3Laboratoire National de Référence, Hôpital Peltier, Ministère de la Santé, Djibouti ville 98230, Djibouti; 4Caisse Nationale de Sécurité Sociale (CNSS), Djibouti ville 98230, Djiboutikahiyamed@gmail.com (K.M.A.);; 5Laboratoire de Diagnostic, Centre de Santé Communautaire d’Einguela, Ministère de la Santé, Djibouti ville 98230, Djibouti; 6Programme National de Lutte Contre le Paludisme, Ministère de la Santé, Djibouti ville 98230, Djibouti; samatark24@gmail.com; 7Service de Santé des Armées, Présidence de la République, Djibouti ville 98230, Djibouti; conseiller.sante@presidence.dj; 8UNDP Djibouti, Global Fund to Fight AIDS-TB-Malaria, Djibouti ville 98230, Djibouti; bouh.abdi@gmail.com; 9Unité Parasitologie et Entomologie, Département Microbiologie et Maladies Infectieuses, Institut de Recherche Biomédicale des Armées, 13005 Marseille, France

**Keywords:** diagnosis, diagnostic performance, Djibouti, histidine-rich protein, lactate dehydrogenase, *Plasmodium falciparum*, *Plasmodium vivax*, rapid diagnostic test

## Abstract

Until 2020, Djiboutian health authorities relied on histidine-rich protein-2 (HRP2)-based rapid diagnostic tests (RDTs) to establish the diagnosis of *Plasmodium falciparum*. The rapid spread of *P. falciparum* histidine-rich protein-2 and -3 (*pfhrp2/3*) gene-deleted parasite strains in Djibouti has led the authorities to switch from HRP2-based RDTs to lactate dehydrogenase (LDH)-based RDTs targeting the plasmodial lactate dehydrogenase (pLDH) specific for *P. falciparum* and *P. vivax* (RapiGEN BIOCREDIT Malaria Ag Pf/Pv pLDH/pLDH) in 2021. This study was conducted with the primary objective of evaluating the diagnostic performance of this alternative RDT. Operational constraints related, in particular, to the implementation of this RDT during the COVID-19 pandemic were also considered. The performance of BIOCREDIT Malaria Ag Pf/Pv (pLDH/pLDH) RDT was also compared to our previously published data on the performance of two HRP2-based RDTs deployed in Djibouti in 2018–2020. The diagnosis of 350 febrile patients with suspected malaria in Djibouti city was established using two batches of RapiGEN BIOCREDIT Malaria Ag Pf/Pv (pLDH/pLDH) RDT over a two-year period (2022 and 2023) and confirmed by real-time quantitative polymerase chain reaction. The sensitivity and specificity for the detection of *P. falciparum* were 88.2% and 100%, respectively. For *P. vivax*, the sensitivity was 86.7% and the specificity was 100%. Re-training and closer supervision of the technicians between 2022 and 2023 have led to an increased sensitivity to detect *P. falciparum* (69.8% in 2022 versus 88.2% in 2023; *p* < 0.01). The receiver operating characteristic curve analysis highlighted a better performance in the diagnosis of *P. falciparum* with pLDH-based RDTs compared with previous HRP2-based RDTs. In Djibouti, where *pfhrp2*-deleted strains are rapidly gaining ground, LDH-based RDTs seem to be more suitable for diagnosing *P. falciparum* than HRP2-based RDTs. Awareness-raising and training for technical staff have also been beneficial.

## 1. Introduction

Malaria is a major public health problem in Djibouti. In recent years, with more than 73,000 malaria cases reported in Djibouti in 2020, malaria has become the leading cause of hospital visits in the country [1]. According to the latest World Health Organisation (WHO) World Malaria Report, Djibouti is one of the countries where the malaria burden has increased [2].

Although malaria diagnosis using molecular methods could provide a robust and accurate assessment of the malaria situation in Djibouti, financial, technical, and logistical constraints restrict its implementation to reference or research laboratories, and it is not currently routinely available at the point of care. Hence, as in many other African countries, malaria diagnosis in Djibouti is mainly based on two laboratory techniques: microscopy and rapid diagnostic test (RDT), as recommended by the WHO. Microscopy still remains the reference method for malaria diagnosis in Africa, as well as in other endemic areas, despite its limitations, including requirements for well-maintained equipment, reagents of analytical quality (methanol, Giemsa stain), electricity, and good technical skills that are not always available in resource-limited settings [2,3,4]. One of the major shortcomings of microscopic examination is encountered in samples with low parasitaemia [2,3]. In addition to light microscopy, RDTs have become an essential tool for the diagnosis of malaria as they do not require high technical and infrastructural resources and can establish a reliable diagnosis within 15 to 20 min [5,6].

RDTs are immunochromatographic assays based on a reaction between specific antibodies fixed to a solid support and parasite antigens or enzymes circulating in the peripheral blood of a patient. The currently available RDTs are designed to detect histidine-rich protein-2 (HRP2) specific for *Plasmodium falciparum*, lactate dehydrogenase (LDH), or aldolase [7]. Each of the four human *Plasmodium* spp. (i.e., *P. falciparum*, *Plasmodium vivax*, *Plasmodium ovale* [with two sub-species], and *Plasmodium malariae*) has a distinct isomer of plasmodial LDH (pLDH), allowing the detection of all human *Plasmodium* spp. using the genus-conserved pLDH sequences (usually referred to as “Pan”) or the individual detection of *P. falciparum* and *P. vivax* using *P. falciparum*-specific pLDH and *P. vivax*-specific pLDH, respectively [8]. Aldolase is common to all four human *Plasmodium* spp., allowing the detection of the genus *Plasmodium*.

Among RDTs, CareStart™ Pf/Pv (*P. falciparum* HRP2/*P. vivax*-specific pLDH) Combo test was evaluated by the WHO in the late 2000s, and its high performance was validated [9]. This RDT has been field-tested in different regions of Ethiopia and has been shown to be reliable in several studies [10,11,12,13]. However, the performance of CareStart™ Pf/Pv Combo, which had been widely deployed throughout the country in Djibouti until 2020, was reported to be poor, especially in detecting the presence of *P. falciparum*, even in patients with signs and symptoms often associated with acute malarial attacks and cured after a presumptive treatment with artemisinin-based combination therapy (ACT) [14]. For this reason, some health centres replaced CareStart™ Pf/Pv Combo test with the Biosynex Pf/Pv (HRP2/Pan pLDH) test in 2020–2021 in an attempt to obtain a higher sensitivity in detecting both *P. falciparum* and *P. vivax*. However, it turned out that field diagnosis of *P. falciparum* using the Biosynex RDT also performed poorly [14]. Based on these observations and feedback from the field, it has been hypothesised that the principal underlying reason for the poor performance of these HRP2-based RDTs in Djibouti may be due to an increasing number of *P. falciparum* strains with deleted *P. falciparum* histidine-rich protein 2 and 3 (*pfhrp2*/*pfhrp3*) genes [15]. In view of the rapidly changing malaria situation in the country, WHO recommended the replacement of CareStart™ Pf/Pv Combo RDT (also Biosynex Pf/Pv test) with a new RapiGEN BIOCREDIT Malaria Ag Pf/Pv (pLDH/pLDH) RDT, which detects *P. falciparum*-specific pLDH and *P. vivax*-specific pLDH in two different bands [16]. Its use started in April 2021, and RapiGEN BIOCREDIT Malaria Ag Pf/Pv (pLDH/pLDH) RDT is currently being deployed in all health services in Djibouti. Faced with the challenge of declining performance of PfHRP2-based RDT in Djibouti, the objective of the present study was to evaluate the performance of the new LDH-based RapiGEN BIOCREDIT Malaria Ag Pf/Pv (pLDH/pLDH) RDT in the field compared to real-time PCR as the gold standard in order to better adjust the country’s malaria control interventions.

## 2. Materials and Methods

### 2.1. Study Population and Diagnostic Screening

Febrile patients of all ages with suspected malaria were screened for malaria using RapiGEN BIOCREDIT Malaria Ag Pf/Pv (pLDH/pLDH) RDT (product reference: C61RHA25; batch number H016A003DA in 2022 and H016B001D in 2023; RapiGEN, Inc.; Suwon, Republic of Korea; hereafter referred to as “BIOCREDIT Pf/Pv (pLDH/pLDH) RDT or test”) in five community health centres (Centre de soin 2 de la Caisse Nationale de Sécurité Sociale de Djibouti, Centre d’Ambouli, Centre Khor Bourhan, Centre Ibrahim Balala, and Centre Arnaud) and a hospital (Hôpital Cheiko) during the transmission period from April to May 2022 and in six community health centres (Centre Farah-Had, Centre Einguella, Centre Khor Bourhan, Centre Ambouli, Centre PK12, Centre Hayableh) and Hospital Balbala in April 2023 in Djibouti city (Figure 1) [14]. A sub-sample was then randomly selected from the suspected malaria cases, regardless of the RDT result. In order to improve RDT reading, the health personnel were re-trained in the use of RDT prior to sample collection in 2023. Fingerpricked blood samples were obtained to perform RDT according to the manufacturer’s instructions. The results were read within 15 min. All RDT results were re-read by a second experienced staff member. The procurement of BIOCREDIT Pf/Pv (pLDH/pLDH) RDT was funded by the Global Fund. The RDT was distributed by the Djiboutian Ministry of Health to community health centres and hospitals and was used for free diagnostic screening for patients. Microscopic examination of blood smears was not performed in the present study due to inadequate training and/or a shortage of laboratory technicians in the health centres where the study was conducted.

### 2.2. DNA Extraction and PCR Diagnosis

Both positive and negative RDT cassettes were collected from the participating health centres and hospitals daily and stored at ambient temperature for less than 6 months until molecular analysis. Parasite DNA was extracted from the proximal portion of the nitrocellulose membrane of the RDT, as previously described by Cnops et al. [17]. DNA was extracted using an automated magnetic bead-based robot (MagMAX™-Express, Thermo Fisher Scientific, Montigny-le-Bretonneux, France) according to the manufacturer’s recommendations.

Real-time quantitative PCR (qPCR) was performed to detect the presence of *P. falciparum* and/or *P. vivax* using two sets of primers developed by Demas et al. [18]. The reaction mixture was prepared with a Type-it high-resolution melting (HRM) PCR kit (Qiagen GmbH, Hilden, Germany) containing EvaGreen^®^ fluorescent DNA intercalating dye (Biotium, Inc., Fremont, CA, USA). The reaction mixture (total volume, 25 µL) consisted of 3 µL of DNA template, 12.5 µL of HRM master mix 2X, 0.12 µL of primers at 150 µM, and 9.26 µL of nuclease-free water. Separate reactions were performed for *P. falciparum* and *P. vivax* with the following two successive cycle parameters developed by Demas et al. [18]: initial step consisting of denaturation at 95 °C for 5 min and 65.5 °C (for *P. falciparum*) or 58 °C (for *P. vivax*) for 30 s, followed by 20 cycles of 95 °C for 30 s and 65.5 °C (for *P. falciparum*) or 58 °C (for *P. vivax*) for 1 min, then a final extension at 72 °C for 1 min. After the amplification cycles were completed, the melt curve was analysed for quality control. The relative quantity of parasite DNA present in the samples was estimated by the quantification cycle (Cq), also called the threshold cycle (Ct). PCR amplifications were performed in 96-well PCR plates on a Bio-Rad CFX96 Touch real-time PCR detection system (Bio-Rad, Marnes-la-Coquette, France). Melt curve and Cq analyses were performed using CFX Maestro software version 2.3 (Bio-Rad, Marnes-la-Coquette, France) for CFX real-time PCR instruments (Bio-Rad).

### 2.3. Statistical Analysis

The RDT results were compared to qPCR results, which were the gold standard of the present study. The sensitivity, specificity, positive predictive value (PPV), and negative predictive value (NPV) were calculated using MedCalc, version 22.018 (MedCalc Software, Ostend, Belgium, https://www.medcalc.org). Cohen’s kappa coefficient was calculated to estimate the degree of agreement between PCR and RDT results using R software version 4.1.2 [19]. The interpretation of the kappa coefficient was classified as follows: <0, no agreement, 0.01–0.20, slight agreement, 0.21–0.40, fair agreement, 0.41–0.60, moderate agreement, 0.61–0.80, substantial agreement, and 0.81–1.0, almost perfect agreement [20]. A comparison of the results obtained in 2022 and 2023, without and with re-reading of RDTs, was carried out using Fisher’s exact test.

The Cq (or Ct) is defined as the number of PCR cycles required for the fluorescent signal to be detectable, i.e., when it attains and exceeds the threshold or background fluorescent level [21]. The fluorescence emission determining the Cq value is inversely proportional to the initial quantity of nucleic acid template present in the sample. A relatively low Cq level (i.e., <29) is indicative of the abundant presence of the target nucleic acid. Cq values 30–40 usually indicate low to moderate quantities of target nucleic acid, suggesting low parasitaemia in our study. In the present study, samples with a Cq value >39 were considered to be negative. The mean Cq values of false negative and true positive RDTs were compared using the unpaired Mann–Whitney test to evaluate whether the quantities of DNA templates in these two sample groups were similar. The significance level was fixed at *p* < 0.05.

For the evaluation of diagnostic test performance, the receiver operating characteristic (ROC) curve was plotted to graphically represent the diseased and non-diseased populations [22]. The curve is plotted with the true positive rate (sensitivity) of a test on the ordinate and the false positive rate (1—specificity) on the abscissa. The ROC curve allows (i) to assess the performance of the diagnostic test, (ii) to define the optimal validity threshold of the test, and (iii) to compare two diagnostic tests. The analysis of the ROC curve is based on the determination of the area under the curve (AUC), also specifically known as the area under the ROC curve (AUROC). The AUROC is a measure of the diagnostic accuracy of a test. If the ROC curve coincides with the diagonal line, the AUROC is 0.5, which corresponds to a test with no diagnostic value. The higher the AUROC, the better the test. Thus, this procedure allows several tests to be compared simultaneously.

In the present study, two additional ROC curves derived from data of two HRP2- and/or pLDH-based RDTs analysed in our earlier study [14] were plotted to compare with the new BIOCREDIT Pf/Pv (pLDH/pLDH) RDT. Three AUROC values, each representing the performance data obtained with different RDTs (CareStart™ RDT Ag Combo RDT (ACCESS BIO; Somerset, NJ, USA), Biosynex RDT (Biosynex; Illkirch-Graffenstaden, France) and BIOCREDIT Pf/Pv (pLDH/pLDH) RDT) used in Djibouti between 2018 and 2023, were compared using a non-parametric test developed by DeLong et al. [23].

## 3. Results

### 3.1. Prevalence of Plasmodia

A total of 350 BIOCREDIT Pf/Pv (pLDH/pLDH) RDTs were collected from nine study sites in Djibouti city, of which 221 (63.1%) were negative and 129 (36.9%) were positive (Table 1). A total of 200 (57.1%) samples had either no gene amplification (i.e., absence of fluorescence) or showed a weak fluorescence signal only after 39 cycles and were considered qPCR-negative (Table 1). qPCR confirmed that 150 of 350 (42.8%) samples were positive, of which 121/150 (80.7%) were infected with *P. falciparum*, 21/150 (14%) with *P. vivax*, and 8/150 (5.3%) with a mixed *P. falciparum*/*P. vivax* infection (Table 1).

### 3.2. Performance of BIOCREDIT Pf/Pv (pLDH/pLDH) RDT

The performance of BIOCREDIT Pf/Pv (pLDH/pLDH) RDT with respect to qPCR as the reference method is summarised in Table 2.

#### 3.2.1. Diagnostic Performance of BIOCREDIT Pf/Pv (pLDH/pLDH) RDT to Detect *P. falciparum*

BIOCREDIT Pf/Pv (pLDH/pLDH) RDT detected *P. falciparum* in 30.6% (107/350) of the samples tested. During the 2-year period of the study, the percentages of RDT-positive *P. falciparum* were 20.2% in 2022 and 43.8% in 2023 (Appendix A). For *P. falciparum*, a total of 25 false negatives were found (8.1% and 5.9% in 2022 and 2023, respectively) (Table 2). False positives (3/198; 1.5%) were observed in 2022 but not in 2023. The sensitivity of the RDT to detect *P. falciparum* was 69.8% (95% CI, 55.7–81.7%) in 2022 and 88.2% (95% CI, 78.7–94.4%) in 2023, respectively. For both years combined, sensitivity was 80.2% [95% CI, 72.7–81.7%].The specificity was slightly better in 2023 (100%) than in 2022 (97.9%; 95% CI, 94.1–99.6%). The difference in the sensitivity for the detection of *P. falciparum* with BIOCREDIT Pf/Pv (pLDH/pLDH) RDT between 2022 and 2023 was statistically significant (*p* < 0.01; Fisher’s exact test). There was no significant difference (*p* > 0.05; Fisher’s exact test) between 2022 and 2023 for the specificity to detect *P. falciparum*.

#### 3.2.2. Diagnostic Performance of BIOCREDIT Pf/Pv (pLDH/pLDH) RDT to Detect *P. vivax*

For *P. vivax*, 2 of 152 (1.3%) samples were false negative in 2023; none was false negative in 2022 (*n* = 198). There was no false-positive result in both years (2022 and 2023). The sensitivity was 100% and 86.7% (95% IC, 59.5%–98.3%) in 2022 and 2023, respectively, but the difference did not reach statistical significance (*p* > 0.05). By contrast, the specificity was 100% in both years. The results of RDT and qPCR were in substantial agreement (kappa coefficient, 0.82; 95% CI, 0.76–0.88).

### 3.3. Comparison of the Performance of HRP2- and pLDH-Based RDTs

The AUROC determined from the results of BIOCREDIT Pf/Pv (pLDH/pLDH) RDT was 0.912 [95% CI, 0.864–0.959], which is considered a relatively good diagnostic test (Figure 2). We found that the AUROC curve of BIOCREDIT Pf/Pv (pLDH/pLDH) RDT compared to that of CareStart™ and Biosynex™ RDTs is significantly different (*p* < 0.05). The AUROC curves of Biosynex™ RDT and CareStart™ RDT were also significantly different (*p* < 0.05).

### 3.4. Relative Quantification of Plasmodial DNA

There was a statistically significant difference (*p* = 0.0002) between the mean Cq value of samples that were false negative (RDT+/qPCR+; 36.0 ± 3.5; *n* = 16) and that of samples that yielded true positive results (RDT−/qPCR+; 26.3 ± 3.2; *n* = 37) in 2022 (Figure 3). The results obtained in 2023 also showed a statistically significant difference (*p* = 0.0002) between the mean Cq value of the false negative samples (34.6 ± 2.7; *n* = 37) and that of the samples with true positive results (29.6 ± 2.9; *n* = 9).

## 4. Discussion

The utility of HRP2-based RDTs for malaria diagnosis is threatened and may be more and more limited by the spread of *P. falciparum* strains with deleted *pfhrp2/3* genes in the Horn of Africa. The advent of anti-malarial treatments administered only after confirmation of malaria infection by RDT obviously exacerbates this trend [24,25,26]. Indeed, two recent studies conducted in Djibouti have shown high proportions (63–83%) of *pfhrp2/3* gene deletion that was directly responsible for false-negative RDT results [15,27]. These data are also consistent with the poor accuracy of HRP2-based RDT recently assessed in Djibouti City [14]. In addition, similar studies conducted in the neighbouring countries in the Horn of Africa (i.e., Ethiopia and Eritrea) also revealed high proportions (7.6–22%) of *pfhrp2/3* gene deletions [28,29]. All of these data led the Djiboutian authorities to deploy a new rapid malaria diagnostic test based on the detection of the LDH protein. The use of HRP2-based RDTs is no longer recommended, considering that above 5% of strains deleted, the proportion of false-negative HRP2-based RDTs due to diagnostic escape would exceed the proportion of false-negative LDH-based RDTs due to reduced sensitivity [30]. Our study aimed to assess the effectiveness of the recently implemented LDH-based RDT in the Djiboutian context.

### 4.1. P. falciparum

Over the two years of our study, LDH-based RDT showed a sensitivity of 80.2% [95% CI, 72.7–81.7%] in the detection of *P. falciparum* in Djibouti city. Comparison with the two HRP2-based RDTs previously used shows a clear improvement versus CareStart™ Pf/Pv (HRP2/pLDH) RDT evaluated in 2018–2020 (40.3%, *p* < 0.05) and more moderate but non-significant improvement versus Biosynex P.f/Pan (HRP2/Pan-pLDH) RDT assessed in 2021 (73.1%, *p* = 0.12) [14]. Moreover, analysis of the ROC curves (Figure 2) shows that this specific LDH-based RDT is more accurate than the two previously evaluated HRP2-based RDTs in Djibouti city. These results confirm the benefit obtained by switching to LDH-based RDTs in the Djiboutian context. However, considering the performances over the two years of the study, the sensitivity of the test is not optimal.

Other studies on the performance of this RDT conducted in Uganda have found higher sensitivity (i.e., 87.8% and 95.8%) than in Djibouti [31,32]. The sensitivity of RDT based on the detection of pLDH is known to vary from one study to another. In some studies, the sensitivity was very high (>95%), while in others, the sensitivity was much lower (<80%) [33]. The lower sensitivity observed in our study is mainly due to a high proportion of false negatives (19.4%).

Many factors could explain this relatively high rate of false negativity of BIOCREDIT Pf/Pv (pLDH/pLDH) RDT. These factors include (i) heat stability, which decreases significantly after storage above 35 °C and 45 °C for similar RDT (product reference: C60RHA25) [34], (as these temperatures are usual in Djibouti, heat could explain the lower sensitivity observed in our study), (ii) manufacturing procedures leading to variations in the quality of different batches [35], (iii) previous unreported use of anti-malarial drugs [36], and (iv) low parasitaemia, which is known to be correlated with poor RDT performance [8,37,38]. Indeed, a similar Pf/Pv (pLDH/pLDH) RDT (product reference C60RHA25) from the same supplier has a panel detection score of 75% for a density of 200 *P. falciparum*/µL [34], and a decrease in sensitivity for low parasitaemia is expected.

Although parasitaemia was not determined by microscopy in the present study, Cq values provide relevant information on parasitaemia in the sample [39]. Our data showed that false-negative samples (RDT−/PCR+) were associated with high Cq values (*p* < 0.05) (Figure 3). This significant association strongly suggests that the primary reason for false-negative RDT results in 16 (2022) and 9 (2023) *P. falciparum* samples in our study was due to low parasitaemia, i.e., below the threshold of detection of LDH-based RDT.

Furthermore, the improvement in sensitivity between the two years highlights the probable contribution of an additional factor. The LDH-based RDT (BIOCREDIT Pf/Pv (pLDH/pLDH) test) sensitivity to detect *P. falciparum* in Djibouti was 69.8% and 88.2% in 2022 and 2023, respectively. This substantial and significant improvement between the two years (*p* < 0.05) is likely to be due to the re-training of technicians in 2023 to correctly use RDT and interpret the result. Better storage conditions for RDTs in 2023 are also possible, given this re-training, although this has not been demonstrated. It is, therefore, reasonable to consider the RDT’s performance measures in 2023 to be less biased than in 2022.

By contrast, only a few cases of false-positive samples (3/350, 0.8%) were found in the present study. All three cases occurred in *P. falciparum*/*P. vivax* mixed infections. These findings may suggest that the patients initially had *P. falciparum* infection, with or without *P. vivax*, but recovered from *P. falciparum* infection, most probably after treatment with ACT [36]. The residual circulating *P. falciparum* pLDH enzyme may have been responsible for the false-positive test result [40]. Even without residual pLDH, false-positive HRP2- or pLDH-based RDT may occur in patients with rheumatoid factor [41]. In the present study, the clinical records of the patients were not available, and this possibility cannot be ruled out. Alternatively, a false-positive result may be due to a cross-reaction between *P. falciparum* pLDH and *P. vivax* pLDH. This latter hypothesis needs further studies for confirmation. The specificity of BIOCREDIT Pf/Pv (pLDH/pLDH) RDT to detect *P. falciparum* was high (98%) in the present study, in agreement with the results of several other studies [31,32,34,42,43].

### 4.2. Plasmodium vivax

The sensitivity and specificity of BIOCREDIT Pf/Pv (pLDH/pLDH) RDT to detect *P. vivax* were high, 93.1% and 100%, respectively, in the present study and consistent with previous reports [31,34]. In contrast to the relatively low sensitivity of BIOCREDIT Pf/Pv (pLDH/pLDH) RDT to detect *P. falciparum*, this RDT was highly sensitive to detect *P. vivax*. Further analysis of our data published earlier showed significant differences (*p* < 0.05) in the sensitivity to detect *P. vivax* between BIOCREDIT Pf/Pv (pLDH/pLDH) RDT (93.1%) and either CareStart™ (71.9%) or Biosynex (65.7%) RDT [14]. As *P. vivax* is the second most common malaria parasite in Djibouti after *P. falciparum*, the BIOCREDIT Pf/Pv (pLDH/pLDH) test is potentially useful for malaria diagnosis in the country.

This is the first assessment of BIOCREDIT Pf/Pv (pLDH/pLDH) RDT that is currently deployed nationwide in Djibouti in an effort to counter the problem of high rates of false-negative *P. falciparum* using PfHRP2-based RDT. The limitations of the present study include a relatively small sample size, in particular for *P. vivax* and *P. falciparum*-*P. vivax* mixed infection, the lack of comparison between pLDH-based and HRP2-based RDT for the same samples, and the absence of molecular analysis on the presence or absence of *pfhrp2/3* gene deletions in our samples with *P. falciparum*. Although microscopic examination of blood smears was not performed to determine parasitaemia, Cq values provide a relative but reliable quantification of parasite DNA and reflect relative parasitaemia between different samples [39], allowing us to deduce that low *P. falciparum* parasitaemia was probably the primary reason for the false-negative result.

## 5. Conclusions

In the current Djiboutian context characterised by a limited technical capacity for microscopy and a high prevalence of *pfhrp2*-deleted *P. falciparum* strains, taking into account the ROC curves, the present study suggests that species-specific pLDH-based BIOCREDIT Pf/Pv (pLDH/pLDH) RDT performed better to detect *P. falciparum* than two previous RDTs based on HRP2 and pan pLDH deployed throughout the country. The current alarming situation in the country that is experiencing a rapid increase in malaria incidence may further deteriorate if the use of HRP2-based RDTs is maintained, as it favours the selection of *pfhrp2*-deleted *P. falciparum* strains and increases the prevalence of these strains. Moreover, the significant prevalence of *P. vivax* in Djibouti, as confirmed in a recent epidemiology study [14], also calls for the deployment of diagnostic tools that are adapted to detect this malaria species. In this context, BIOCREDIT Pf/Pv (pLDH/pLDH) RDT and other RDTs based on the separate detection of *P. falciparum*- and *P. vivax*-specific pLDH might be the best option for the diagnosis of malaria in Djibouti today. However, difficulties in detecting the lowest parasitaemia levels and heat stability raise serious concerns. Further studies should be conducted on a larger number of samples collected from other sentinel sites in the country to further assess the performance of BIOCREDIT Pf/Pv (pLDH/pLDH) RDT and evaluate its effectiveness country-wide and in mixed infections.

## Figures and Tables

**Figure 1 diagnostics-14-00262-f001:**
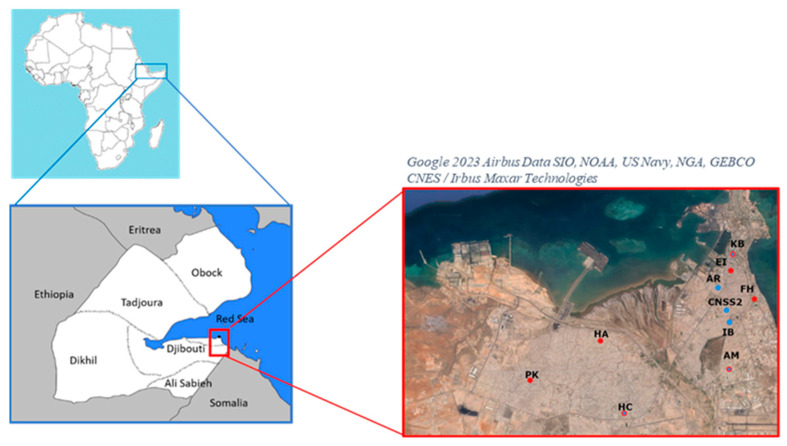
Map of Djibouti with a focus on inclusion health centres in 2022–2023. Two recruitment campaigns were carried out in Djibouti city during 2022 (blue circles) in the community health centres of Arnaud (AR) and Ibrahim Balala (IB) and the social security care centre 2 (CNSS2) and during 2023 (red circles) in the community health centres of Einguella (EI) and PK12 (PK) and the polyclinic centres of Farah-Had (FH) and Hayableh (HA). The community health centres from Ambouli (AM), Khor Bourhan (KB), and Cheiko Hospital (HC) were involved during the two years of recruitment (hatched red and blue circles).

**Figure 2 diagnostics-14-00262-f002:**
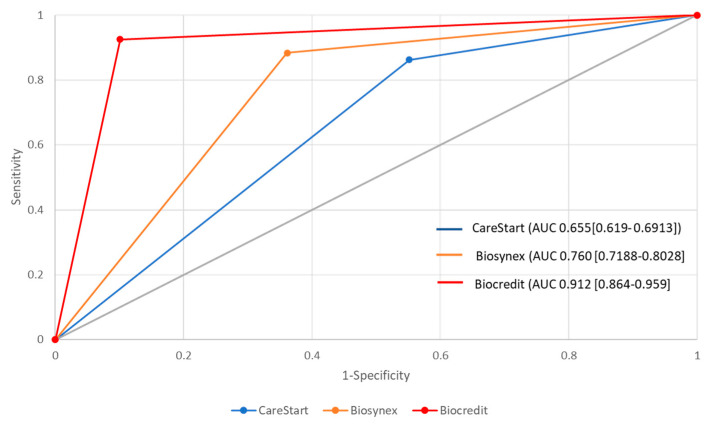
ROC curves of RDTs. The area under receiving operating characteristic (AUROC) curves of three RDTs evaluated in the present study (BIOCREDIT Pf/Pv (pLDH/pLDH) RDT; red line) and our previous study (CareStart Malaria™ Pf/Pv HRP2/pLDH RDT, Access Bio Inc., Somerset, NJ, USA, the blue line, and Biosynex^®^ Pf/Pan RDT, Biosynex, Illkirch-Graffenstaden, France, the yellow line) [14]. AUC is the area under the curve. The 95% confidence interval is presented in brackets.

**Figure 3 diagnostics-14-00262-f003:**
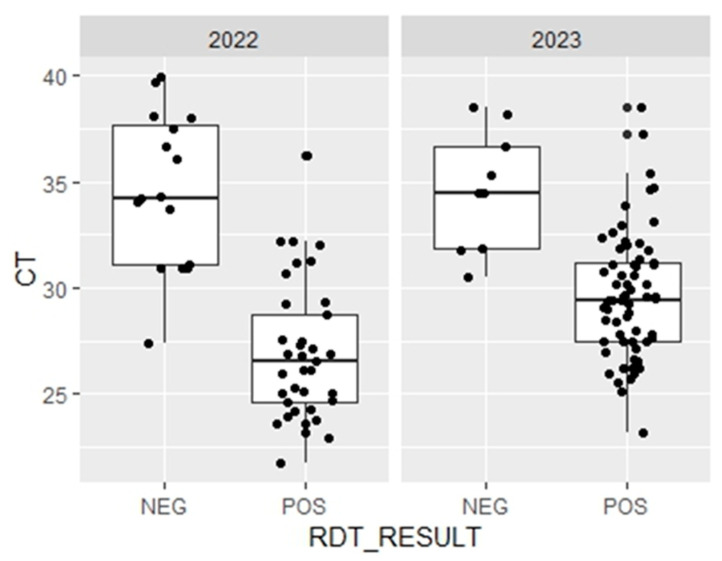
Comparison of threshold cycle (Ct) between false negative (RDT_LDH_−/PCR+) and true positive (RDT_LDH_+/PCR+) in Djibouti (2022–2023). The threshold cycle (Ct) values, considered as a proxy for parasitaemia, are presented for each diagnosis according to the results of the RDT POS (RDT_LDH_+/PCR+) and NEG (RDT_LDH_−/PCR+, i.e., false negatives) and by year. The mean Ct for false negatives was significantly higher than that for true positives based on the unpaired Mann–Whitney test (*p* < 0.05).

**Table 1 diagnostics-14-00262-t001:** Malaria diagnosis by BIOCREDIT Pf/Pv (pLDH/pLDH) rapid diagnostic test and real-time quantitative PCR in patients with suspected malaria in Djibouti city.

Diagnostic Result	*n* (%)	
	RDT	qPCR
*P. falciparum*	102 (29.2)	121 (34.6)
*P. vivax*	22 (6.3)	21 (6.0)
*P. falciparum*/*P. vivax* mixed infection	5 (1.4)	8 (2.3)
Negative *	221 (63.1)	200 (57.1)
Total	350 (100)	350 (100)

*n*, number of samples; RDT, rapid diagnostic test; qPCR, real-time quantitative polymerase chain reaction. * Negative qPCR test denotes a quantification cycle (Cq) value of ≥39 in the present study. The denominator to calculate the proportions of different *Plasmodium* spp. detected by RDT and qPCR was 350 in this table.

**Table 2 diagnostics-14-00262-t002:** Diagnostic performance of BIOCREDIT Pf/Pv (pLDH/pLDH) rapid diagnostic test compared to real-time quantitative PCR in Djibouti city, 2022–2023.

Performance	Year			
	2022		2023	
	*n*	% [95% CI]	*n*	% [95% CI]
*P. falciparum*				
True positive	37	18.7	67	44.1
False positive	3	1.5	0	0
True negative	142	72.0	76	50.0
False negative	16	8.1	9	5.9
Total	198	100	152	100
Sensitivity	–	69.8 [55.7–81.7]	–	88.2 * [78.7–94.4]
Specificity	–	97.8 [93.7–99.5]	–	100 [95.3–100]
PPV	–	92.5 [79.9–97.5]	–	100 [94.6–100]
NPV	–	89.3 [84.6–92.6]	–	89.4 [80.8–95.0]
Accuracy	–	89.9 [84.7–93.8]	–	94.1 [89.1–97.3]
*P. vivax*				
True positive	14	7.0	13	8.6
False positive	0	0	0	0
True negative	184	93.0	137	90.1
False negative	0	0	2	1.3
Total	198	100	152	100.0
Sensitivity	–	100 [76.8–100]	–	86.7 [59.5–98.3]
Specificity	–	100 [98.0–100]	–	100 [97.3–100]
PPV	–	100.0	–	100 [75.3–100]
NPV	–	100.0	–	98.6 [94.9–99.8]
Accuracy	–	100 [98.1–100]	–	98.7 [95.3–99.8]

*n*, number of samples; 95% CI, 95% confidence interval; PPV, positive predictive value; NPV, negative predictive value. * Significant *p*-values < 0.01 for the comparison of two different batches of BIOCREDIT Pf/Pv (pLDH/pLDH) RDT between 2022 and 2023 after raising awareness and updating training for laboratory staff.

## Data Availability

The data presented in this study are available on request from the corresponding author. The data on RapiGEN BIOCREDIT Pf/Pv (pLDH/pLDH) RDT are not publicly available due to the confidentiality of patient data. The data on the performance of Biosynex RDT and CareStart RDT were published in our earlier work [14].

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
