# Peer review of "Assessment of the Performance of Lactate Dehydrogenase-Based Rapid Diagnostic Test for Malaria in Djibouti in 2022–2023"

_diagnostics, 2024, doi:10.3390/diagnostics14030262_

Round 1

Reviewer 1 Report

Comments and Suggestions for Authors

Overall this is an good study with interesting findings. It's well written and presented. The topic is important. I appreciated the Discussion about the reasons for continuing high numbers of false negatives even with the new RDTs.

A few areas where explanation of results could be improved:

1. Please be consistent in how you refer to the new test. Sometimes Biocredit (e.g. Fig 2), sometimes RapidGen (e.g. title to Table 1). it took me a little time to realize they were the same thing.

2. please enlarge the map in right of Fig 1. It is impossible to see where the superimposed red and blue circles are, and hard to see the red and blue ones

3. lines 124 to 129. please make clear that DNA extraction was attempted from all RDT strips if that was the case. At first I thought it was only from positives.

4. Line 211 mentions supplementary data but I don't see any with the paper

5. Where do the percentages of 70% and 60% in the abstract line 33-34 come from? I don't see them easily in the main paper results

6. Line 171 - I think it should say "1 minus specificity" (false positive rate) not just specificity

7.line 218 - what test was used to detemine no significant difference between 2022 and 2023? more info needed.

8. Line 343. Is malaria really increasing 'exponentially'? That has a specific meaning and I don't think so .  Perhaps rapid or alarming increase would be better.

Author Response

Overall this is an good study with interesting findings. It's well written and presented. The topic is important. I appreciated the Discussion about the reasons for continuing high numbers of false negatives even with the new RDTs.

A few areas where explanation of results could be improved:

  1. Please be consistent in how you refer to the new test. Sometimes Biocredit (e.g. Fig 2), sometimes RapidGen (e.g. title to Table 1). it took me a little time to realize they were the same thing.

Authors’ response: We apologize for the confusion. Although the complete commercial name is RapiGEN BIOCREDIT Malaria Ag Pf/Pv (pLDH/pLDH) rapid diagnostic test, in the WHO publication (Malaria rapid diagnostic test performance: summary results of WHO product testing of malaria RDTs: round 1-8 (2008-2018), this RDT is referred to as “BIOCREDIT Malaria Ag Pf/Pv (pLDH/pLDH)” without “RapiGEN.” In fact, “RapiGEN” is the name of the manufacturer (RapiGEN Inc.). For further clarity, we mentioned the complete name in the abstract, in the Background section, and the first time the RDT was mentioned in the Methods section, and thereafter this RDT was referred to as BIOCREDIT Malaria Ag Pf/Pv (pLDH/pLDH) RDT or test, as we mentioned in line 99. It is indeed important for the readers not to confuse this BIOCREDIT RDT with other available BIOCREDIT RDT (HRP-2 or pan LDH). “BIOCREDIT” was also written in capital letters throughout the text for consistency.

We have also included the reference number of this RDT from its supplier in the materials and methods section to avoid confusion for readers. The WHO-FIND report evaluated the product C60RHA25, whereas we used the product C61RHA25 in the field. Although these two products appear to be very similar in that they target the same Pf/Pv (pLDH/pLDH) antigens, differences are also likely. The panel detection scores presented by the supplier suggest an improvement in performance, but no data is available describing the evaluation protocol. These elements are considered in the discussion section.

  1. please enlarge the map in right of Fig 1. It is impossible to see where the superimposed red and blue circles are, and hard to see the red and blue ones

Authors’ response: To take these comments into account, we have enlarged the circles to improve their visibility. The superimposed circles, which are difficult to see, have been replaced by hatched circles (blue/red). We hope that these changes will improve the visibility of the figure. The editor can also enlarge the image according to pagination constraints if required.

  1. lines 124 to 129. please make clear that DNA extraction was attempted from all RDT strips if that was the case. At first I thought it was only from positives.

Authors’ response: As stated in the Results section (lines 187-188), all 350 RDTs that were available, positive or negative, were analyzed. This information was added in the Methods section for further clarity: “Both positive and negative RDT cassettes were collected.”

  1. Line 211 mentions supplementary data but I don't see any with the paper

Authors’ response: Table S1 was initially inserted into the text in the "Supplementary Materials" section. In line with the reviewer's recommendations, this table has been moved to a separate file entitled Table S1.

  1. Where do the percentages of 70% and 60% in the abstract line 33-34 come from? I don't see them easily in the main paper results

Authors’ response: We agree that these values were ambiguous. The 70% was an approximation of the sensitivity of the diagnosis of Plasmodium falciparum by RDT (69.8%) in 2022. The abstract has been amended to clarify this. The 60% corresponds to the efficacy of HRP2-based rapid diagnostic tests in our previous study. As suggested by the reviewers, this result should not appear in the abstract, so we have removed it.

  1. Line 171 - I think it should say "1 minus specificity" (false positive rate) not just specificity

Authors’ response: Thank you for pointing out our mistake. In fact, in Fig. 2 (ROC curves) the X-axis legend was given correctly as “1 – specificity.” The statement was revised as follows: “The curve is plotted with the true positive rate (sensitivity) of a test on the ordinate and the false positive rate (1 – specificity) on the abscissa.”

7.line 218 - what test was used to detemine no significant difference between 2022 and 2023? more info needed.

Authors’ response: We mentioned in lines 155-156 that “a comparison of the results obtained in 2022 and 2023 was carried out using Fisher’s exact test.” We added the statistical test after the P value, in the results section.

  1. Line 343. Is malaria really increasing 'exponentially'? That has a specific meaning and I don't think so.  Perhaps rapid or alarming increase would be better.

Authors’ response: We agree that, from a mathematical viewpoint, an exponential increase refers to a function f(x) = ex, which does not apply to the context described in line 343. As suggested, we replaced the term “exponentially” with “rapid” (and not “alarming” because the sentence starts with “the current alarming situation.”

Reviewer 2 Report

Comments and Suggestions for Authors

Diagnostics/diagnostics-2738141-peer-review-comments-v1

Assessment of the Performance of Lactate  Dehydrogenase-Based Rapid Diagnostic Test for Malaria in Djibouti in 2022–2023

Rahma Abdi Moussa et.al.

Overall Comments :

Malaria is a significant public health concern in the Horn of Africa, where misdiagnosis of P. falciparum using HRP-based RDTs has become an increasingly worrying issue. As per WMR 2022 between September 2021 and September 2022, investigations of Pfhrp2/3 deletions were reported in 17 publications from 17 countries: Benin, Brazil, Cameroon, the Democratic Republic of the Congo, Djibouti, Ecuador, Equatorial Guinea, Eritrea, Ethiopia, Gabon, Ghana, India, Kenya, Madagascar, Rwanda, Sierra Leone and the United Republic of Tanzania. Of these, only Equatorial Guinea, Kenya and Rwanda did not identify any Pfhrp2 deletions, although deletions in these three countries have been reported in previous publications.

The paper by Rahma Abdi Moussa et.al. is an important contribution to the field, as it evaluates the performance of the new LDH-based RapiGEN Biocredit Pf/Pv (pLDH/pLDH) RDT compared to real-time PCR as the gold standard in Djibouti. The study was conducted on 350 febrile patients of all ages with suspected malaria in Djibouti city using two batches of this new RDT over two years (2022 and 2023). All RDT results were re-read by trained staff and the staff was re-trained in the use of RDTs before sample collection in 2023 to improve RDT reading. A comparison of the results obtained in 2022 and 2023, without and with re-reading of RDTs, was also done. RDT cassettes were collected from the participating health centres and hospitals daily and stored at ambient temperature for less than 6 months until molecular analysis.

While the paper is well-written, there seems to be some confusion regarding the methodology and interpretation of results vis-à-vis the objective of the study, and minor revisions and corrections are suggested under 'Specific Comments' before publication of the paper.

Specific Comments :

Abstract:  The abstract appears confusing due to the following:

Results: The objective of the study is quite clear but I find the results unclear. Given the fact that the objective of the study was to evaluate the diagnostic performance of the LDH-based RDTs for Pf and Pv in comparison to qPCR(Gold standard for this study), results should be specific to the objective. The sensitivity and specificity based on operational issues related to RDT performance by the health workers cannot be a part of the diagnostic performance. This can be added after the results are given.  In case you feel that the difference in sensitivity is related to the difference in batches, then revise accordingly giving the batch numbers used during 2022 and 2023. The methodology in the main text might also need revision accordingly. By saying the sensitivity improved in 2023, it's not possible to interpret how it improved. Please revisit your observations and while doing so consider the WHO-FIND RDT performance results for this RDT also.

Conclusion: It is not clear how you concluded that LDH-based RDTs are more sensitive (70% versus 60%) in diagnosing Plasmodium falciparum than HRP2-based RDTs despite reduced sensitivity for detecting low parasitaemia. This field study did not compare the performance of the LDH-based RDTs with the previously used HRP-based RDTs. The conclusion should, therefore, report the performance of the LDH-based RDTs used in this study.

In case the comparison with other RDTs used in previous studies based on the AUROC is intended to be included here, it may be added separately to avoid confusion and misinterpretation.

Consider revision considering the above facts.

Introduction :

Line 44-46: I found ambiguity and confusion in this line as the paper refers to the WMR 2022(Ref 2). Please review in view of the following excerpts from WMR 2022 :

·        WMR 2022: Page no. 22 & 23 :

 3.4 ESTIMATED MALARIA CASES AND DEATHS IN THE WHO EASTERN MEDITERRANEAN REGION, 2000–2021 :

'Between 2020 and 2021, increases in estimated malaria cases were seen in Somalia, the Sudan and Yemen, with an additional 205 000, 64 000 and 180 000 cases, respectively. Estimated cases reduced in Afghanistan, Djibouti and Pakistan’. Estimated malaria deaths also reduced by about 45%, from 13 600 in 2000 to 7500 in 2014, and then increased by 79% between 2014 and 2021 to reach 13 400 deaths (Table 3.4). This increase in deaths was due to increases in Djibouti, Somalia, the Sudan and Yemen. Mortality rates have gradually increased since 2016 by a total of 28%. In 2021, the Sudan accounted for most of the estimated malaria cases in this region (54%), followed by Somalia, Yemen, Pakistan, Afghanistan and Djibouti (Fig. 3.5c).

In 2021, all countries in the region reported zero malaria deaths apart from Djibouti, the Sudan and Yemen.

 8.4 WHO EASTERN MEDITERRANEAN REGION :

Since 2015, there has been an increase in case incidence and mortality rates in the WHO Eastern Mediterranean Region, and these rates are now off track by 60% and 65%, respectively (Fig. 8.6). Djibouti, the Sudan and Yemen were off track, with malaria case incidence higher by 40% or more. Malaria mortality rates decreased by 40% or more in Afghanistan and Pakistan in 2021 compared with 2015. There was no increase or decrease in Somalia, but deaths increased by 40% or more in Djibouti, the Sudan and Yemen.

Line 47-49: WHO recommends RDTs or microscopy for routine diagnosis of malaria throughout the world. Therefore, making a case for molecular methods for routine diagnosis at the point of care does not seem to be appropriate here. The available molecular methods also require highly skilled staff, laboratory, equipment and many more facilities. Also, Reference 3 doesn’t appear to be relevant to this statement, being a methods manual on microscopy. It is better to cite the reference of a published paper. Updated WHO guidelines and manuals are also now available on microscopy which can serve as better reference material, if required.

Please consider revising this appropriately from a practical point of view.

 Materials and Methods :

The paper shows improvement in diagnostic performance of RDTs in 2023 over 2022. Consider adding the batch numbers of RTDs used during 2022 and 2023 since discussion related to batches has been included later in the paper. If different operational reasons like lack of training contributed to the difference, please mention how these were measured/quantified. Adding supporting data later in the result, if relevant, should be considered.

Line 105-106: Why was the need for retraining felt, since all RDT results were re-read by a second experienced staff member? Were discrepancies in the results noticed? If yes, how were they addressed?

 What were the patient selection criteria?  How was the decision on inclusion and exclusion of patients taken? What was the denominator(suspected febrile malaria cases) for each area from which the sample was chosen?

 Line 155-156: A comparison of the results obtained in 2022 and  2023, without and with re-reading of RDTs, was carried out using Fisher’s exact test. Why was this required? Please add a few lines to justify this point. Was untrained staff utilized during 2022?

Line 208-226: Please revisit the headings and revise for clarity to ensure that the intended meaning is conveyed.

Line 228-232: CareStart™  and Biosynex RDTs were not used in this study. The methodology and conditions in the previous studies are expected to be different. What is the relevance of comparing the AUROC curves of the HRP-based RDTs not used here with Biocredit™? What are the likely biases to be introduced by doing so? Please comment while interpreting and discussing the results.

Discussion :

Line 272-274: Please add references since the comparison was not done in the present study.

Line 297-302: The methodology clearly states that all RDT results were read by a second person. Also, these RDTs were used for DNA extraction. The authors as such should be in a position to comment with confidence on the performance of the RDTs since the number of patients tested is quite limited. Any study wherein the performance of the RDT is evaluated is expected to utilize well trained staff in the first place. Why was retraining required? Please comment on these aspects also.

Abbreviations: Please review the format/style used and present appropriately for better understanding of readers.

Data Availability Statement and Acknowledgements :

Please add relevant information under these headings. At present the instructions to authors have been mentioned.

References :

Check the references again for correctness as per journal format and delete duplications e.g., year in Reference no. 1& 2 & 42.

Author Response

Overall Comments:

Malaria is a significant public health concern in the Horn of Africa, where misdiagnosis of P. falciparum using HRP-based RDTs has become an increasingly worrying issue. As per WMR 2022 between September 2021 and September 2022, investigations of Pfhrp2/3 deletions were reported in 17 publications from 17 countries: Benin, Brazil, Cameroon, the Democratic Republic of the Congo, Djibouti, Ecuador, Equatorial Guinea, Eritrea, Ethiopia, Gabon, Ghana, India, Kenya, Madagascar, Rwanda, Sierra Leone and the United Republic of Tanzania. Of these, only Equatorial Guinea, Kenya and Rwanda did not identify any Pfhrp2 deletions, although deletions in these three countries have been reported in previous publications.

The paper by Rahma Abdi Moussa et.al. is an important contribution to the field, as it evaluates the performance of the new LDH-based RapiGEN Biocredit Pf/Pv (pLDH/pLDH) RDT compared to real-time PCR as the gold standard in Djibouti. The study was conducted on 350 febrile patients of all ages with suspected malaria in Djibouti city using two batches of this new RDT over two years (2022 and 2023). All RDT results were re-read by trained staff and the staff was re-trained in the use of RDTs before sample collection in 2023 to improve RDT reading. A comparison of the results obtained in 2022 and 2023, without and with re-reading of RDTs, was also done. RDT cassettes were collected from the participating health centres and hospitals daily and stored at ambient temperature for less than 6 months until molecular analysis.

While the paper is well-written, there seems to be some confusion regarding the methodology and interpretation of results vis-à-vis the objective of the study, and minor revisions and corrections are suggested under 'Specific Comments' before publication of the paper.

Specific Comments :

Abstract:  

The abstract appears confusing due to the following:

Results: The objective of the study is quite clear but I find the results unclear. Given the fact that the objective of the study was to evaluate the diagnostic performance of the LDH-based RDTs for Pf and Pv in comparison to qPCR (Gold standard for this study), results should be specific to the objective. The sensitivity and specificity based on operational issues related to RDT performance by the health workers cannot be a part of the diagnostic performance. This can be added after the results are given.  In case you feel that the difference in sensitivity is related to the difference in batches, then revise accordingly giving the batch numbers used during 2022 and 2023. The methodology in the main text might also need revision accordingly. By saying the sensitivity improved in 2023, it's not possible to interpret how it improved. Please revisit your observations and while doing so consider the WHO-FIND RDT performance results for this RDT also.

Authors’ response: We agree with the reviewer that our objectives need to be clarified, particularly in the abstract. As stated in the introduction (lines 92-94) “the objective of the present study was to evaluate the performance of the new LDH-based RapiGEN Biocredit Pf/Pv (pLDH/pLDH) RDT in the field compared to real-time PCR as the gold standard in order to better adjust the country's malaria control interventions”. The operational aspect of our study, adapted to the Djiboutian context, justifies the evaluation of the RDT under real conditions with laboratory staff dedicated to malaria diagnosis before, during and after our study in Djibouti. This is particularly important in a country where the local skills available for malaria diagnosis are a limiting factor.

The quality controls implemented are expected to limit the impact of human failure in our study. Nevertheless, we were unable to avoid them entirely. In addition, the introduction of a new test during the COVID-19 pandemic may also have been an aggravating factor, which could have resulted in inappropriate laboratory practices. The corrective actions implemented in 2023 with updated training for laboratory staff were accompanied by a significant improvement in the sensitivity of the RDT. This more recent evaluation was also accompanied by a change in the batch number of the RDT used, but our protocol does not allow us to distinguish the contribution of these two parameters. However, we have noted an improvement in the diagnosis of malaria.  

In following the recommendations of the reviewer, we have presented this important result, specifying how it could bias the evaluation of the performance of the RDT (lines 339-346), the evaluation of which is our primary objective. The values for 2023 are therefore presented as the least biased evaluation of the performance of malaria diagnosis by RDT compared with those for 2022 (lines 346-347). Only values for 2023 appear in the abstract. The improvement between 2022 and 2023 is mentioned in the abstract, but the detailed values are only presented in the discussion section to avoid confusion for readers.

Although our protocol does not allow us to assess the contribution of the batch to the improved diagnostic performance by RDT observed in 2023, batch number may be useful for a meta-analysis and has been added in the section material and method (line 113-114).

We would like to thank the reviewer for suggesting the additional reference (WHO-FIND RDT performance evaluation program), which we believe to be highly relevant. Two major results of this study relating to the Biocredit Pf/Pv (pLDH/pLDH) RDT have been added in the discussion section: the moderate detection score for low parasitaemia (75% for 200 P. falciparum/µl) (lines 329-332) and the poor heat stability after storage at temperatures of 35°C and 45°C (lines 324-327). The corresponding reference has been added (reference 34). However, we would point out to the reviewer that the product reference evaluated in this report (C60RHA25) does not correspond exactly to the reference used in Djibouti during our study (61RHA25). These products are undoubtedly very similar, but we have no factual evidence to support this. Significant improvements may also have been made, but the latest panel detection scores, available from the supplier, do not specify the conditions under which they were obtained independently of the WHO-FIND program. References to this report must therefore take this into account. We have therefore added the product reference of the RDT used in our study (lines 113-114 ) and that of the RDT evaluated in the WHO-FIND program (lines 326, 331).

Conclusion: It is not clear how you concluded that LDH-based RDTs are more sensitive (70% versus 60%) in diagnosing Plasmodium falciparum than HRP2-based RDTs despite reduced sensitivity for detecting low parasitaemia. This field study did not compare the performance of the LDH-based RDTs with the previously used HRP-based RDTs. The conclusion should, therefore, report the performance of the LDH-based RDTs used in this study.

In case the comparison with other RDTs used in previous studies based on the AUROC is intended to be included here, it may be added separately to avoid confusion and misinterpretation.

Authors’ response: Part of this comment regarding 70% vs 60% is similar to the other reviewer’s comment. We agree that the presentation of these results is confusing. The 70% is an approximation of the sensitivity of Plasmodium falciparum diagnosis by LDH-based RDT and 60% refers to previous studies on HRP2-based RDT. As requested by the reviewers we have removed these percentages from the abstract and we emphasized our findings on the performance of the LDH-based RDTs (our primary objective).

The comparison of the performance of the LDH-based RapiGEN Biocredit Pf/Pv (pLDH/pLDH) RDT with two previously studied HRP2-based RDTs deployed in Djibouti was redefined as a secondary objective in the abstract. The result of this comparison appears at the end of the abstract to avoid possible confusion.

Consider revision considering the above facts.

Introduction:

Line 44-46: I found ambiguity and confusion in this line as the paper refers to the WMR 2022(Ref 2). Please review in view of the following excerpts from WMR 2022 :

WMR 2022: Page no. 22 & 23:

3.4 ESTIMATED MALARIA CASES AND DEATHS IN THE WHO EASTERN MEDITERRANEAN REGION, 2000–2021 :

'Between 2020 and 2021, increases in estimated malaria cases were seen in Somalia, the Sudan and Yemen, with an additional 205 000, 64 000 and 180 000 cases, respectively. Estimated cases reduced in Afghanistan, Djibouti and Pakistan’. Estimated malaria deaths also reduced by about 45%, from 13 600 in 2000 to 7500 in 2014, and then increased by 79% between 2014 and 2021 to reach 13 400 deaths (Table 3.4). This increase in deaths was due to increases in Djibouti, Somalia, the Sudan and Yemen. Mortality rates have gradually increased since 2016 by a total of 28%. In 2021, the Sudan accounted for most of the estimated malaria cases in this region (54%), followed by Somalia, Yemen, Pakistan, Afghanistan and Djibouti (Fig. 3.5c).

In 2021, all countries in the region reported zero malaria deaths apart from Djibouti, the Sudan and Yemen.

8.4 WHO EASTERN MEDITERRANEAN REGION :

Since 2015, there has been an increase in case incidence and mortality rates in the WHO Eastern Mediterranean Region, and these rates are now off track by 60% and 65%, respectively (Fig. 8.6). Djibouti, the Sudan and Yemen were off track, with malaria case incidence higher by 40% or more. Malaria mortality rates decreased by 40% or more in Afghanistan and Pakistan in 2021 compared with 2015. There was no increase or decrease in Somalia, but deaths increased by 40% or more in Djibouti, the Sudan and Yemen.

Authors’ response: The reviewer is comparing the malaria situation in Djibouti to the rest of the WHO Eastern Mediterranean region. In this context, we agree with the reviewer that our statement is not true (“in sharp contrast to many endemic countries where malaria prevalence has decreased or remained stable”). We were actually comparing the malaria situation in Djibouti to that of some countries in the WHO African region. The WHO WMR 2022 supports and confirms our statement that “Djibouti is one of the countries where malaria burden has increased” (line 44-45). We are deleting “in sharp contrast to many endemic countries where malaria prevalence has decreased or remained stable” because this statement is not clear without naming individual African countries.

Line 47-49: WHO recommends RDTs or microscopy for routine diagnosis of malaria throughout the world. Therefore, making a case for molecular methods for routine diagnosis at the point of care does not seem to be appropriate here. The available molecular methods also require highly skilled staff, laboratory, equipment and many more facilities. Also, Reference 3 doesn’t appear to be relevant to this statement, being a methods manual on microscopy. It is better to cite the reference of a published paper. Updated WHO guidelines and manuals are also now available on microscopy which can serve as better reference material, if required.

Authors’ response: We agree with the reviewer. It’s a misunderstanding. In fact, what we say in lines 47-49 (“several constraints restrict molecular methods to reference or research laboratories; it is not available at the point of care”) supports the reviewer’s viewpoint. Other statements made in the same paragraph, in lines 52-59 (“Microscopy still remains the reference method for malaria diagnosis” “RDTs have become an essential tool for the diagnosis of malaria”) further support the reviewer’s (and our) viewpoint. To make our point clearer, we added “as recommended by the WHO” to line 52 and revised the following statement in line 53: Microscopy still remains the reference method for malaria diagnosis in Africa “as well as in other endemic areas.”

As for Reference 3, we would like to point out that, first of all, both Reference 3 and 4 were cited in line 55, Ref 3 to support our (and WHO’s) position that “malaria microscopy remains a major reference standard for field trials of clinical interventions and other diagnostic platforms” (citation from ref 3, page 9, the opening statement of the document) and Ref 4 to support the statement that RDT is an alternative to microscopy. In the revised manuscript, we also added Ref 2, which refers to the “existing point-of-care platforms (specifically, RDTs and microscopy)” and describes the limits of microscopy and currently available RDTs in pages 108-109 of Ref 2.

Please consider revising this appropriately from a practical point of view.

Materials and Methods:

The paper shows improvement in diagnostic performance of RDTs in 2023 over 2022. Consider adding the batch numbers of RTDs used during 2022 and 2023 since discussion related to batches has been included later in the paper. If different operational reasons like lack of training contributed to the difference, please mention how these were measured/quantified. Adding supporting data later in the result, if relevant, should be considered.

Authors’ response: As suggested, the batch numbers were added. The effect of training in 2023, compared to 2022, is difficult to assess objectively. Our protocol is not designed to identify the different contributions of operational reasons in improving malaria diagnosis between these two years. In fact, the updating of laboratory staff training was triggered by two events after the start of the study. The first one was RDT reading errors observed after the deployment of a new pan-malaria test in 2021 in Djibouti city. These initial results from previous studies were not available until after the current study had begun. The second event was occasional feedback from the field suggesting deviations from good laboratory practice in the use of RDTs. These two events prompted an update of the training courses, but did not lead to a quantification of the problem. However, an overall contribution from operational reasons (i.e. batch variation, staff training) can be assessed indirectly by comparing the RDT results with gold standard in 2022 and 2023. This procedure was performed by statistical comparison of the results, as described in lines 155-156.

Line 105-106: Why was the need for retraining felt, since all RDT results were re-read by a second experienced staff member? Were discrepancies in the results noticed? If yes, how were they addressed?

Authors’ response: The reviewer raises this point several times in his/her comments. Our full explanation is given in our replies to the reviewer’s comments on Lines 155-156 and Lines 297-302 below. Please consider also our answer to the previous question.

 What were the patient selection criteria?  How was the decision on inclusion and exclusion of patients taken? What was the denominator (suspected febrile malaria cases) for each area from which the sample was chosen?

Authors’ response: We mentioned in line 98 that “febrile patients of all ages with suspected malaria were screened for malaria.” A sub-sample was then randomly selected from the suspected malaria cases regardless of the RDT result. This information has been added to the corresponding materials and methods section (lines 121-122).

 Line 155-156: A comparison of the results obtained in 2022 and 2023, without and with re-reading of RDTs, was carried out using Fisher’s exact test. Why was this required? Please add a few lines to justify this point. Was untrained staff utilized during 2022?

Authors’ response: Technicians were not retrained before the study in 2022, but they were previously trained for the study conducted in 2018-2021. We suspect that COVID-19 pandemics have disorganized Djiboutian health system as in many countries (World Malaria Report 2022). Moreover, successive changes of RDTs with different interpretations (i.e. species-specific or pan-malaria results) have probably contributed to possible misreadings.

In addition to our reply given here, we replied to the reviewer’s comments on lines 297-302: In the field, particularly in the African context, we have learned from our experience that when some of the technicians are not working under close supervision of their more experienced superiors, errors in procedures can occur. A regular retraining of technicians is an important component of malaria control strategy.

Line 208-226: Please revisit the headings and revise for clarity to ensure that the intended meaning is conveyed.

Authors’ response: The titles of the sub-sections 3.2, 3.2.1, 3.2.2, and 3.3 were revised.

Line 228-232: CareStart™  and Biosynex RDTs were not used in this study. The methodology and conditions in the previous studies are expected to be different. What is the relevance of comparing the AUROC curves of the HRP-based RDTs not used here with Biocredit™? What are the likely biases to be introduced by doing so? Please comment while interpreting and discussing the results.

Authors’ response: While we agree with the reviewer’s comments, from the practical viewpoint, the Djiboutian Ministry of Health could not have introduced two or three different RDTs simultaneously in the country for evaluation. In our study, the evaluation of three RDTs was performed in such a way that the methodology, including the inclusion criteria of malaria-infected patients, as well as the study sites (all performed in Djibouti city) and transmission season, would be similar during the entire study period (2018-2023). We agree that biases would inevitably be introduced, especially the possible increase in the number of pfhrp2-deleted parasites which were undetected during the period when only HRP2-based RDTs were used. We mentioned these limitations of the study at the end of the discussion. We think that within the limits of this study, AUROC curves summarize and illustrate pretty well the performance of each RDT used in the country.

Discussion:

Line 272-274: Please add references since the comparison was not done in the present study.

Authors’ response: Thank you for your reminder. We added the reference [14].

Line 297-302: The methodology clearly states that all RDT results were read by a second person. Also, these RDTs were used for DNA extraction. The authors as such should be in a position to comment with confidence on the performance of the RDTs since the number of patients tested is quite limited. Any study wherein the performance of the RDT is evaluated is expected to utilize well trained staff in the first place. Why was retraining required? Please comment on these aspects also.

Authors’ response: We agree with the reviewer’s viewpoint. However, in the field, particularly in the African context, we have learned from our experience that when some of the technicians are not working under close supervision of their more experienced superiors, errors in procedures can occur. A regular retraining of technicians is an important component of malaria control strategy. The two events that led to the updating of staff training mentioned in the previous questions confirm this. In addition, the effect of the COVID-19 pandemic may also have had a negative impact.

Abbreviations: Please review the format/style used and present appropriately for better understanding of readers.

Authors’ response: We rechecked the abbreviations used in the text, including the list of abbreviations given at the end of the main text. The abbreviations in the abstract were explained. The abbreviation (RDT) in the keywords was deleted.  

Data Availability Statement and Acknowledgements :

Please add relevant information under these headings. At present the instructions to authors have been mentioned.

Authors’ response: We thank the reviewer for his vigilance and apologise for this omission. We have added the corresponding statements.

References:

Check the references again for correctness as per journal format and delete duplications e.g., year in Reference no. 1& 2 & 42.

Authors’ response: We rechecked all the cited references and reformatted them according to the format recommended by the journal. The years in references 1, 2, and 42 are not duplications. The year is part of the article title.